# Automatic Ore Blending Optimization Algorithm for Sintering Based on the Cartesian Product

Xinying Ren [1,2,3], Chaoyi Gao [4], Hanchen Wang [5], Shilong Feng [1,2,3], Tao Xue [2,6] and Aimin Yang [1,2,3,6,*]

1. College of Metallurgy and Energy, North China University of Science and Technology, Tangshan 063210, China
2. Hebei Intelligent Engineering Research Center of Iron Ore Optimization and Ironmaking Raw Materials Preparation Process, North China University of Science and Technology, Tangshan 063210, China
3. The Key Laboratory of Engineering Computing in Tangshan City, North China University of Science and Technology, Tangshan 063210, China
4. College of Management, North China University of Science and Technology, Tangshan 063210, China
5. Yisheng College, North China University of Science and Technology, Tangshan 063210, China
6. College of Science, North China University of Science and Technology, Tangshan 063210, China
* Correspondence: aimin@ncst.edu.cn

**Abstract:** In actual sinter production, batching is a complex metallurgical and mathematical problem. Aiming at the problem of the precising batching of iron ore in the process of sintering batching, an automatic batching algorithm based on a Cartesian product to batch sinter was proposed for the first time. When the algorithm is applied to the sintering batching process, a complete batching scheme can be obtained, which can realize the organic combination with other calculation processes, can effectively save the manpower and material cost of sintering batching, and is of great significance to the comprehensive use of iron ore resources. Taking the actual sintering production batching of a domestic iron and steel plant as an example, according to the batching requirements compared with various ore batching schemes, combined with the actual production situation, the automatic batching optimization algorithm based on a Cartesian product is applied to build a mathematical model of sintering batching. Through the algorithm test, the practicability of the automatic batching algorithm is verified. In addition, the automatic batching algorithm based on a Cartesian product has good performance in other batching fields.

**Keywords:** mixed ingredients; Cartesian product; sinter; sintering batching



## 1. Introduction

In the 1970s, due to the few types of iron ore used in sintering, a simple theoretical method was generally used to realize sintering batching in iron and steel production. Since the 1980s, researchers have applied the linear programming method to sintering batching. At first, the optimization object was limited to the chemical composition of sinter. With the deepening of research content, the physical and metallurgical properties of sintering have also become the optimization objectives of the model.

The linear programming method requires that the objective function and constraints of the problem to be solved are linear. Although there are many successful examples of applying the linear programming method, the linear programming method still has some shortcomings: with the increase in the number of constraints and the increase of the optimization model, there will be no solution. At this time, it is necessary to repeatedly adjust the constraints or even delete the constraints to obtain the solution, which leads to a reduction in the ease of use of the method.

Based on the above analysis, it can be seen that the traditional batching method heavily depends on the operator's experience and there are many problems faced in multi-objective methods, multimineral types, nonlinear optimization, and so on. Therefore, with the development of artificial intelligence technology, foreign countries began to apply artificial

intelligence to the field of batching in the 1980s [1] and conducted in-depth research on this basis. In recent years, the application of the artificial intelligence algorithm in sintering automatic batching optimization has achieved remarkable results. The algorithms applied to batching include the genetic algorithm, the particle swarm optimization algorithm, the ant colony algorithm, and a mixture of some algorithms (also known as a hybrid algorithm) [2–6].

Using the computer to realize automatic batching has the advantages of making the process fast, stable, and labor saving. Therefore, automatic batching is also widely used in the chemical industry, the agricultural food industry, and other fields. Wang et al. [7] designed and developed a set of automatic real-time batching systems for raw coal and coal gangue to realize the automatic, accurate, and stable proportioning of coal gangue in coal chemical enterprises and applied it to actual production. The results show that the automatic real-time batching system can effectively solve the problem of control accuracy of dynamic metering. It can effectively improve the batching quality, reduce the labor intensity of workers, and improve the production efficiency. To stabilize the quality of cement production and increase the economic benefits of enterprise production, Li [8] tested the application of the cement raw meal batching system for automatic raw meal batching. A long-term application test shows that the cement raw material batching system has a good effect on the control of raw material batching in a cement plant. It can ensure the qualified rate of raw materials and reduce production costs, which has been highly recognized by enterprises. Li et al. [9], combined with the guide to the construction of core competitiveness of feed enterprises, deeply analyzed the automatic batching system applied to feed enterprises. Three key settings that need special attention in the automatic batching system are summarized, the formula operating system, system configuration, and batching control. It is concluded that the reliability and safety of the feed processing and production can be greatly improved by using computer automatic batching. Liu et al. [10] designed an automatic batching system with a high degree of automation in view of the shortcomings of manual batching and tested its accuracy, efficiency, and reliability. It is concluded that the automatic batching system has higher efficiency, lower error, and more stable batching than manual batching. The application of automatic batching system can also improve the working environment, reduce costs, and improve benefits for enterprises.

For the research of automatic batching in the sintering process, many researchers optimize the automatic batching system by upgrading hardware and improving the algorithm. To optimize sintering batching to make it better and faster on site, Li et al. [11] studied the sintering characteristics of four kinds of iron concentrate powders at room temperature and high temperatures, including the balling index, the assimilation temperature, liquid phase fluidity, bonding phase strength, and the calcium ferrite generation capacity, and conducted a sintering batching test on these characteristics to obtain the optimal sintering batching according to the high and low matching of various properties of iron concentrate powder. It can not only effectively reduce the sintering batching cost and process energy consumption but also significantly improve the air permeability in the sintering process and improve the calcium ferrite formation capacity and strength of the sinter. It provides a reference for further optimizing sintering batching and improving sinter quality. Gan et al. [12] realized the detection of the chemical composition of the mixture and timely automatic batching by introducing the online composition detection system. To some extent, it alleviates the problem of lag in the adjustment of sinter chemical composition and promotes the optimization of sinter quality. Li et al. [13] established the corresponding mathematical model by investigating the relationship between iron ore grade, output, and waste emission in the stages of mining, preselection, beneficiation, and smelting pellets; carried out marginal analysis on this basis to obtain the grade link cost relationship model in each stage; and proposed the grade cost marginal cost index considering the waste treatment cost to evaluate the grade index. The integrated optimization method of key grade indexes is put forward. Gu et al. [14] considered that the influencing factors of beneficiation are not fully considered in the polymetallic sintering batching model, resulting in the lack of refinement. A multi-objective sintering batching optimization mathematical model based on the mining and beneficiation process

is established with the optimization objectives of grade deviation, percentage deviation of ore lithology, total output deviation, and mining and transportation cost. Based on the standard genetic algorithm, the mutation process and the comparison selection process of the algorithm are improved and an adaptive genetic algorithm is designed to solve the model. Through research and analysis, the model can ensure the balance of ore grade and ore washability, increase the applicability of the sintering batching optimization model, and bring the sintering batching optimization model more in line with production practice. Feng et al. [6] proposed a constrained multi-objective particle swarm optimization algorithm based on region division to optimize sintering proportioning in view of the impact of large changes in iron ore price, ore grade fluctuation, complex sintering raw material information, and various sintering proportioning constraints on sintering proportioning cost in the sintering proportioning process of iron and steel production. To coordinate the relationship between global exploration and local search, the adaptive angle division strategy is integrated into the constraint evaluation criteria, the local optimal solution information is extracted combined with the regional distribution, and the dual external save set mechanism is introduced to maintain the population diversity. The effectiveness of the proposed algorithm is verified by the test of the standard function set. When the algorithm is applied to the sintering batching process, it can take into account the cost and the total iron content and effectively reduce the cost of sintering proportioning. It is of great significance to the comprehensive utilization of sintered iron ore resources and quality assurance. To reduce the raw material cost of the blast furnace ironmaking system and realize the collaborative optimization of the whole process from iron ore procurement to blast furnace ironmaking, He et al. [15] developed a whole system and whole process optimized sintering batching platform for blast furnace ironmaking. Taking iron ore as the starting point, based on the calculation of blast furnace ironmaking process and material balance, the data analysis and calculation model is established by using the algorithms of planning solution, linear regression, multiple nonlinear regression, and neural network and the optimal sintering batching decision is formed through the statistical analysis of the data.

Most of the existing sintering ore blending optimization research is improved on the basis of the linear programming algorithm or through experimental comparison, the introduction of real-time hardware detection, and so on. The disadvantage is that for the nonlinear sintering batching calculation process, the sintering batching results cannot be obtained. Moreover, it cannot provide multiple sintering batching optimization schemes for users to make the comparison and decision and the program has strong initiative. This paper starts with the raw material proportioning mechanism of sintering automatic batching, develops an automatic batching algorithm based on a Cartesian product, and builds a sintering automatic batching model. In view of the nonlinear sintering batching calculation in the sintering batching process of current enterprises, the sintering batching optimization scheme is recommended and the requirements for the comparison of sinter quality under different raw material ratios are met. Compared with other batching algorithms, the Cartesian product batching model can reduce the calculation times of automatic batching to a certain extent and obtain the batching scheme faster.

## 2. Mathematical Modeling of Sintering Proportioning

### 2.1. Core Requirements of Sintering Ingredients

In the actual sintering work, when adjusting the proportion of raw materials, one only needs to adjust the proportion of several raw materials in a certain range in order to find a suitable sintering batching scheme. When adjusting the raw material ratio, the adjustment accuracy is not high due to the influence of process equipment. For example, when carrying out sintering operation, Shanxi Jianlong Industrial Co., Ltd., first limits the calculable indexes, such as the alkalinity of sinter, and adjusts the proportion of a few materials to obtain a sintering proportioning scheme that meets the requirements. Even if there is a demand for changing materials, it is usually to change an iron-containing raw material and then adjust the proportion of all raw materials. It will not change a

variety of raw materials at the same time. When adjusting the proportion of raw materials, while limiting the basicity and other indicators of sinter, it is often limited by raw material reserves and batching experience. Based on this, each raw material is limited to a range for proportioning and the automatic proportioning is carried out according to a certain allocation step, which can reduce the time complexity of the automatic proportioning algorithm to a certain extent.

Because the kernel function of the automatic batching algorithm uses a Cartesian product, the research on this algorithm comes from the "intelligent sintering batching" project and this algorithm can not only be used for sintering batching research but also be applied to other batching fields with restrictions on raw materials. To sum up, an intelligent batching algorithm based on a Cartesian product (CP-IB) is proposed. The following introduction uses "CP-IB" as the name of the algorithm.

*2.2. CP-IB Mathematical Model*

1. Raw material proportioning range

$$r_i = [a_i \ b_i] \qquad (i = 1, 2, 3 \cdots n) \tag{1}$$

where $r_i$ is the value range of the $i$-th raw material ratio and $b_i$ and $a_i$ correspond to the maximum ratio and the minimum ratio of the $i$-th raw material value, respectively.

2. Possible value of raw material ratio

$$m_i = \left\lfloor \frac{b_i - a_i}{s} \right\rfloor \tag{2}$$

$$R_i = (a_i, \ a_i + s, \ a_i + 2s \ \cdots \ a_i + m_i * s) \qquad (i = 1, 2, 3 \cdots n) \tag{3}$$

where $s$ is the step of raw material proportioning, $s \geq 0$, $m_i$ is the maximum number of steps of the $i$-th raw material proportioning, and $R_i$ is the possible value of the $i$-th raw material ratio.

3. Total number of all raw material combinations

$$z_i = m_i + 1 \tag{4}$$

$$Z = \prod_{i=1}^{n} z_i \tag{5}$$

where $Z_i$ is the number of possible values of the $i$-th raw material ratio and $Z$ is the number of combinations of possible values of all raw material ratios for Cartesian product calculation.

4. A raw material proportioning scheme

$$L_j = \left\{ \left( a_{j1}, \ a_{j2} \cdots a_{jn} \right) \big| \ a_i \in R_i \right\} \qquad (j = 1, 2, 3 \cdots Z, \ i = 1, 2, 3 \cdots n) \tag{6}$$

where $L_j$ is the raw material proportioning scheme of group $j$ and $a_{ji}$ is the ratio of the $i$-th raw material under group j.

5. Proportioning scheme of all raw materials

$$L = (L_1, \ L_2, \ L_3 \cdots L_Z) \qquad (i = 1, 2, 3 \cdots n) \tag{7}$$

$$L = \begin{bmatrix} a_{11} & \cdots & a_{1n} \\ \vdots & \ddots & \vdots \\ a_{Z1} & \cdots & a_{Zn} \end{bmatrix} \tag{8}$$

wherein, in Formula (7), l represents all raw material proportioning scheme groups and the Z line of the matrix represents the proportioning ratio of raw materials

under the *Z* combination. Equation (8) uses different forms to represent *L*, where $a_{ji}$ represents the blending ratio of the *i*-th raw material under the *j*-th blending scheme.

*2.3. Algorithm Implementation and Test*

2.3.1. Algorithm Implementation Process

When the program carries out automatic batching, it needs to calculate to screen out all the ore powder proportioning schemes that meet the limited conditions.

Firstly, the program calculates the value range of all raw materials and stores them. The specific steps are as follows: Firstly, enter the value range of M raw materials [$x_{1i}$ $x_{2i}$], where *i* = 1,2,3...*m*, and set the step size as *n*. Secondly, the values of the *i*-th raw material divided according to the step size are stored in *raw_material_ratio_list* in turn. After all values are stored, the contents of *raw_material_ratio_list* are stored in *all_raw_material_ratio_list* to store the values of the *i*-th raw material. Then, according to the above steps, the value range of m raw materials is successively stored in *all_raw_material_ratio_list* by using the circulation structure, so as to realize the storage of all raw material ranges.

Secondly, the program calculates the proportion scheme of each raw material. Based on the above steps, firstly, calculate the total number of proportioning schemes of m kinds of raw materials and record it as total. Then, for the *i* + 1 batching scheme, take the i/step%lenl value of the L raw material and save the result in *programme_list*. According to the above value rules, traverse m raw materials and store them in *programme_list* successively. After all raw material values are stored, store the value in *programme_list* into *all_programme_list* as the *i* + 1 batching scheme.

Then, according to the above steps, cycle to the total batching scheme, obtain the results of the total batching scheme, and finally store them in *all_programme_list*. The algorithm flow chart is shown in Figure 1.

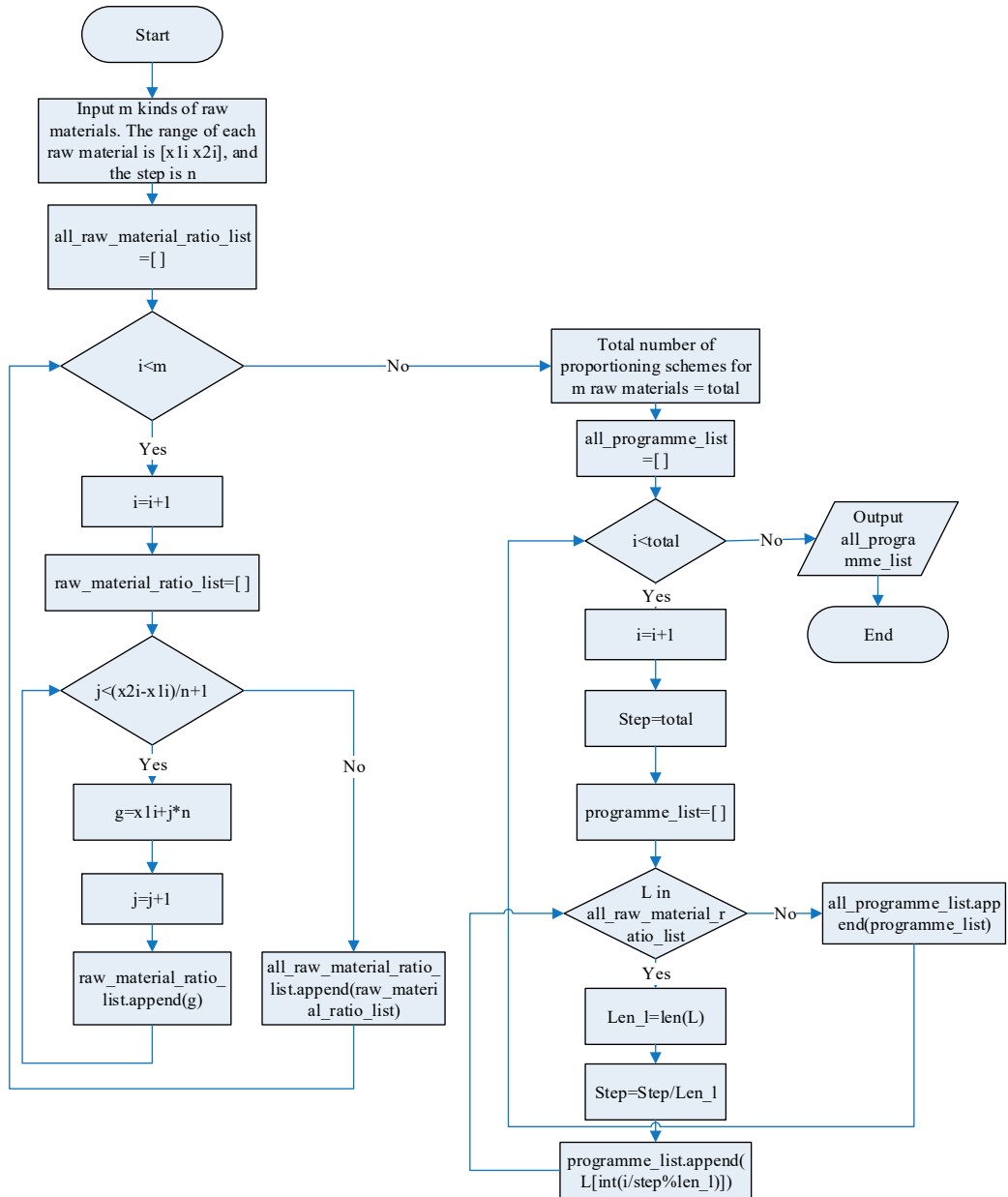

**Figure 1.** Algorithm flow chart.

2.3.2. Algorithm Implementation Pseudocode

The pseudo code of the algorithm is as follows:
Function drange
Begin
list = []
R = Starting point
While (R < End) do
{
list.append(R)
R = R + step
}
return list
End
Function cartesian_product_generator
Begin

```
for i in (Proportioning range of each raw material)
{
for j in drange
{
raw_material_ratio_list.append(j)
}
all_raw_material_ratio_list.append(raw_material_ratio_list)
}
total=sum(len(all_raw_material_ratio_list))
all_programme_list = []
for(i = 0; i < total; i++)
step = total
programme_list = []
for l in all_raw_material_ratio_list
{
len_l = len(l)
step/ = len_l
proportioning=l[i/step%len_l)]
programme_list.append()
}
all_programme_list.append(programme_list)
return all_programme_list
End
```

### 2.3.3. Algorithm Test

The time complexity of the algorithm is usually recorded as:

$$T(k) = Of(k) \tag{9}$$

where $k$ is the data scale, $f(k)$ is the time frequency, and $Of(m)$ is the progressive time complexity of the algorithm. $T(k)$ is a function of the problem scale $k$, which means that with the increase in the problem scale K, the algorithm execution growth rate and time frequency are the same. The above formula is called "time complexity of progressive algorithm," which is called "time complexity" for short.

Generally, when the scale of $k$ data tends to infinity, the algorithm with the slowest growth of $T(k)$ has the shortest execution time and the algorithm is the best.

According to the above definition of time complexity, the algorithm in this paper is tested and evaluated:

The time complexity retains only the highest order of the function. Considering that the algorithm to be tested contains two nested loops, the command under the innermost loop runs the maximum number of times, so the number of executions of this command is selected to represent the time complexity:

$$O(m * s) \tag{10}$$

$$s = \sum_{i=1}^{m} \frac{x_{2i} - x_{1i} + n}{n} \tag{11}$$

where $m$ is the type of sintering batching raw materials, $s$ is the sum of the number of value ranges of all sintering batching raw materials, $x_{2i}$ represents the upper bound of the value of the second sintering batching raw material, $x_{1i}$ represents the lower bound of the value of the $i$-th sintering batching raw material, $n$ represents the step size, and $O(m * s)$ represents the time complexity of the algorithm. It can be seen that the time complexity of the algorithm will increase with the decrease in the step size but the order of the time complexity is relatively low. So the algorithm is better.

Algorithm measurement:

Take r1 = [1, 2], r2 = [3, 3], r3 = [4, 5], and step size s = 0.1. The CP-IB algorithm is tested, and 1331 schemes are obtained. Partial results are shown in Table 1.

**Table 1.** The graph showing the measured results of the algorithm.

| First Set of Results | Second Set of Results | Group III Results |
|---|---|---|
| [1.0, 3.0, 4.0] | [1.0, 3.0, 4.1] | [1.0, 3.0, 4.2] |

Lists [[1, 2], [3, 3], [4, 5]] with a step size of 0.1 were used as input to test the algorithm. The results are shown in Figure 2.

```
All programs:  [[1.0, 3.0, 4.0], [1.0, 3.0, 4.1], [1.0, 3.0, 4.2], [1.0, 3.0, 4.3], [1.0, 3.0, 4.4], [1.0,
 3.0, 4.5], [1.0, 3.0, 4.6], [1.0, 3.0, 4.7], [1.0, 3.0, 4.8], [1.0, 3.0, 4.9], [1.0, 3.0, 5.0], [1.1, 3.0
, 4.0], [1.1, 3.0, 4.1], [1.1, 3.0, 4.2], [1.1, 3.0, 4.3], [1.1, 3.0, 4.4], [1.1, 3.0, 4.5], [1.1, 3.0, 4.
6], [1.1, 3.0, 4.7], [1.1, 3.0, 4.8], [1.1, 3.0, 4.9], [1.1, 3.0, 5.0], [1.2, 3.0, 4.0], [1.2, 3.0, 4.1],
[1.2, 3.0, 4.2], [1.2, 3.0, 4.3], [1.2, 3.0, 4.4], [1.2, 3.0, 4.5], [1.2, 3.0, 4.6], [1.2, 3.0, 4.7], [1.2
, 3.0, 4.8], [1.2, 3.0, 4.9], [1.2, 3.0, 5.0], [1.3, 3.0, 4.0], [1.3, 3.0, 4.1], [1.3, 3.0, 4.2], [1.3, 3.
0, 4.3], [1.3, 3.0, 4.4], [1.3, 3.0, 4.5], [1.3, 3.0, 4.6], [1.3, 3.0, 4.7], [1.3, 3.0, 4.8], [1.3, 3.0, 4
.9], [1.3, 3.0, 5.0], [1.4, 3.0, 4.0], [1.4, 3.0, 4.1], [1.4, 3.0, 4.2], [1.4, 3.0, 4.3], [1.4, 3.0, 4.4],
 [1.4, 3.0, 4.5], [1.4, 3.0, 4.6], [1.4, 3.0, 4.7], [1.4, 3.0, 4.8], [1.4, 3.0, 4.9], [1.4, 3.0, 5.0], [1.
5, 3.0, 4.0], [1.5, 3.0, 4.1], [1.5, 3.0, 4.2], [1.5, 3.0, 4.3], [1.5, 3.0, 4.4], [1.5, 3.0, 4.5], [1.5, 3
.0, 4.6], [1.5, 3.0, 4.7], [1.5, 3.0, 4.8], [1.5, 3.0, 4.9], [1.5, 3.0, 5.0], [1.6, 3.0, 4.0], [1.6, 3.0,
4.1], [1.6, 3.0, 4.2], [1.6, 3.0, 4.3], [1.6, 3.0, 4.4], [1.6, 3.0, 4.5], [1.6, 3.0, 4.6], [1.6, 3.0, 4.7]
, [1.6, 3.0, 4.8], [1.6, 3.0, 4.9], [1.6, 3.0, 5.0], [1.7, 3.0, 4.0], [1.7, 3.0, 4.1], [1.7, 3.0, 4.2], [1
.7, 3.0, 4.3], [1.7, 3.0, 4.4], [1.7, 3.0, 4.5], [1.7, 3.0, 4.6], [1.7, 3.0, 4.7], [1.7, 3.0, 4.8], [1.7,
3.0, 4.9], [1.7, 3.0, 5.0], [1.8, 3.0, 4.0], [1.8, 3.0, 4.1], [1.8, 3.0, 4.2], [1.8, 3.0, 4.3], [1.8, 3.0,
 4.4], [1.8, 3.0, 4.5], [1.8, 3.0, 4.6], [1.8, 3.0, 4.7], [1.8, 3.0, 4.8], [1.8, 3.0, 4.9], [1.8, 3.0, 5.0
], [1.9, 3.0, 4.0], [1.9, 3.0, 4.1], [1.9, 3.0, 4.2], [1.9, 3.0, 4.3], [1.9, 3.0, 4.4], [1.9, 3.0, 4.5], [
1.9, 3.0, 4.6], [1.9, 3.0, 4.7], [1.9, 3.0, 4.8], [1.9, 3.0, 4.9], [1.9, 3.0, 5.0], [2.0, 3.0, 4.0], [2.0,
 3.0, 4.1], [2.0, 3.0, 4.2], [2.0, 3.0, 4.3], [2.0, 3.0, 4.4], [2.0, 3.0, 4.5], [2.0, 3.0, 4.6], [2.0, 3.0
, 4.7], [2.0, 3.0, 4.8], [2.0, 3.0, 4.9], [2.0, 3.0, 5.0]]
```

**Figure 2.** Output image of algorithm test results.

In the above case, the running time of the algorithm is 0.30700 ms. Change the deployment step to 0.05 and keep the rest unchanged. In all, 9261 schemes are obtained. Partial results are shown in Table 2.

**Table 2.** The second measured results of the algorithm.

| First Set of Results | Second Set of Results | Group III Results |
|---|---|---|
| [1.0, 2.0, 4.0] | [1.0, 2.0, 4.05] | [1.0, 2.0, 4.1] |

The running time is 2.04620 ms. Then, to obtain the general variation of running time under different step size conditions, the step size is adjusted to 0.025, 0.0125, 0.00625, and 0.003125, and the running time of the algorithm is shown in Figure 3.

It can be seen from Figure 3 that when the step size is 0.0625, the running time is 61.997 ms, and when the step size is 0.03125, the running time is 245.5 ms. It can be concluded that with a decrease in the step size, the running time of the algorithm increases, but in general, the running time is relatively small and the algorithm effect is good. At the same time, in the actual sintering ore blending production, the accuracy requirements are not high. The general adjustment of the raw material ratio can meet the production requirements when the accuracy reaches 0.1. Therefore, it can be seen that this algorithm can achieve excellent results when applied to the actual sintering ore blending production.

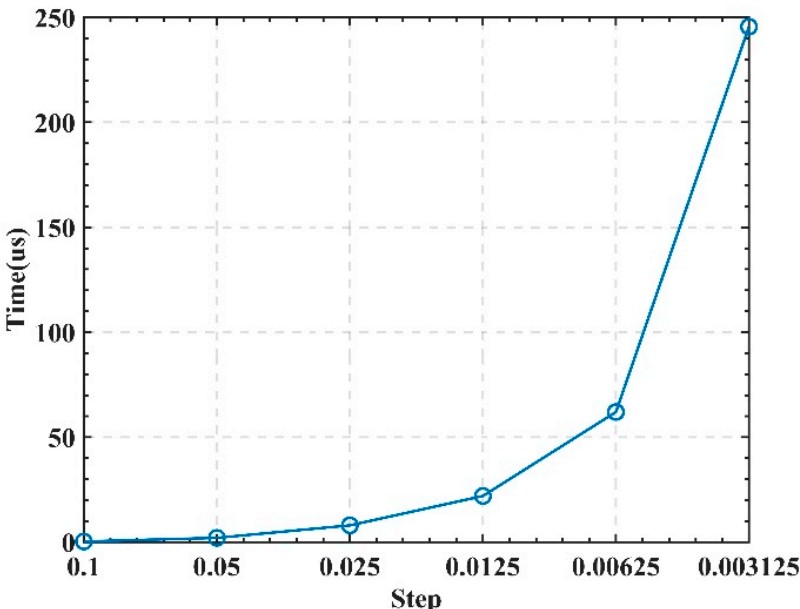

**Figure 3.** Running time of algorithms with different step sizes.

## 3. Application of CP-IB in Sintering Intelligent Ore Blending

### 3.1. Basic Information of Sintering Raw Materials

According to the sinter proportioning data of a sintering plant, the existing available sintering schemes are automatically proportioned by using the CP-IB model and screened according to certain conditions. Sintering raw materials include iron-containing raw materials, fluxes, and fuels, among which, iron-containing raw materials include jinbuba powder, 65 product refined powder, ultra special powder, FMG mixed powder, iron oxide scale, blast furnace dust, high return dust, dust sludge, and magnetic separation powder. Fluxes include rotary kiln lime, dolomite powder, and limestone powder. Fuels include self returning coke powder. The composition of sintering raw materials is shown in Table 1. The content of each component of raw materials is the percentage of the component in the mass of the raw materials. The water content refers to the percentage of the mass of water in the total mass of the raw materials. The burning loss is the sintering loss rate of the raw materials. In addition, there is the cost price of raw materials, which is not listed here because the price will fluctuate over time. The basic properties of sintering raw materials are shown in Table 3.

**Table 3.** Table of basic properties of sintering materials.

| Raw Material Name | Fe (%) | CaO (%) | MgO (%) | SiO₂ (%) | Al₂O₃ (%) | P (%) | S (%) | K₂O (%) | Na₂O (%) | Zn (%) | Pb (%) | Water (%) | Burning Loss |
|---|---|---|---|---|---|---|---|---|---|---|---|---|---|
| Jinbuba | 60.59 | 0.04 | 0.11 | 4.81 | 3.05 | 0.111 | 0.024 | 0 | 0 | 0 | 0 | 7.68 | 5.2 |
| 65 refined | 65.11 | 0.47 | 0.51 | 5.8 | 0.51 | 0.018 | 0.573 | 0.076 | 0.067 | 0.0026 | 0.02 | 10.588 | −1.12 |
| Super special | 57.17 | 0.07 | 0.08 | 6.08 | 3.39 | 0.079 | 0.019 | 0.02 | 0 | 0 | 0.02 | 10.4 | 8.57 |
| FMG mixed | 58.5 | 0.02 | 0.08 | 5.4 | 2.44 | 0.067 | 0.031 | 0.02 | 0 | 0 | 0.02 | 8.4 | 7.68 |
| Iron oxide scale | 68.39 | 1.92 | 0.2 | 2.67 | 0.36 | 0.1 | 0 | 0 | 0 | 0 | 0 | 5 | −2 |
| Blast furnace dust | 53 | 11.41 | 2.46 | 1.23 | 2.6 | 0.13 | 0.049 | 0 | 0 | 0 | 0 | 6 | 11.3 |
| High return | 54.8 | 12.3 | 2.772 | 5.76 | 2.56 | 0.12 | 0.08 | 0 | 0 | 0 | 0 | 3.05 | 0.3 |
| Dust and mud | 55 | 5 | 1.82 | 4 | 1 | 0.1 | 0.06 | 0 | 0 | 0 | 0 | 30 | 4.48 |
| Magnetic separation | 45.55 | 13.5 | 6.74 | 8.417 | 1.527 | 0 | 0 | 0 | 0 | 0 | 0 | 1 | −2 |
| Rotary kiln lime | 0 | 85 | 4.68 | 2.3 | 0.71 | 0.03 | 0.08 | 0 | 0 | 0 | 0 | 0 | 7.8 |
| Dolomite | 0 | 32 | 19.65 | 2 | 0.75 | 0.01 | 0.07 | 0 | 0 | 0 | 0 | 2.12 | 45 |
| Limestone | 0 | 50 | 3.7 | 2.5 | 0.6 | 0.01 | 0.05 | 0 | 0 | 0 | 0 | 1.63 | 44 |
| Self coking | 0 | 0.78 | 0.39 | 7.84 | 5.61 | 0.06 | 0.99 | 0 | 0 | 0 | 0 | 10 | 82.87 |

### 3.1.1. Proportioning Range of Sintering Raw Materials

During sintering production, the proportion of raw materials will be affected by raw material inventory and production cycle. Specifically, in a production cycle, the continuous supply of each raw material selected in the batching scheme and the quality of sintering

should be ensured, which limits the proportion of each raw material. Therefore, the proportion of each raw material can be limited to a range. When changing materials in sintering production, the proportion of some raw materials will not change in the calculation of changing materials because of their stable supply and relatively stable quality. For the above reasons, the proportion range of raw materials in this test is shown in Table 4. Among them, the proportion range refers to the percentage of the quality of a certain raw material in the total quality of all sintering materials. When the upper and lower limits are consistent, it means that the proportion of raw materials has not changed.

**Table 4.** Upper and lower limits of sintering raw material ratio.

| Proportioning Range | Jinbuba | 65 Refined | Super Special | FMG Mixed | Iron Oxide Scale | Blast Furnace Dust | High Return | Dust and Mud | Magnetic Separation | Rotary Kiln Lime | Dolomite | Limestone | Self Coking |
|---|---|---|---|---|---|---|---|---|---|---|---|---|---|
| Lower limit (%) | 12 | 20 | 18 | 15 | 0.7 | 0.8 | 11.5 | 1.5 | 1.23 | 5.5 | 6 | 2 | 4 |
| Upper limit (%) | 14 | 20 | 18 | 15 | 0.7 | 0.8 | 11.5 | 1.5 | 1.23 | 5.5 | 7 | 3 | 4 |

### 3.1.2. Sinter Quality Constraints

To ensure the quality of charging in blast furnace smelting, it is necessary to ensure that the iron grade, alkalinity, and mg Al ratio of sinter meet the production requirements. Therefore, it is necessary to restrict the quality of sinter in order to ensure the stability of blast furnace smelting and the quality of molten iron. In this test, the basicity and the Mg/Al ratio of sinter are limited. At the same time, to improve the quality of sinter, the element content (including Fe, Si, CA, Mg, Al) of sinter is limited. The specific quality constraint range is shown in Table 5.

**Table 5.** Restriction range of sinter quality.

| Quality Constraint Range | Alkalinity | Mg/Al Ratio | Fe | Si | Ca | Mg | Al |
|---|---|---|---|---|---|---|---|
| Lower limit (%) | 1.2 | 0.6 | 0.53 | 0.05 | 0.09 | 0.023 | 0.02 |
| Upper limit (%) | 1.4 | 0.8 | 0.56 | 0.07 | 0.13 | 0.04 | 0.032 |

### 3.2. Model Effect and Analysis

The CP-IB model is applied to sinter proportioning. Using the sintering raw materials, the raw material proportioning range, and constraint parameters on sinter quality in Section 3.1, the raw material proportioning step is set to 0.1 and 87 ore proportioning schemes meeting the requirements are obtained. It includes the batching scheme that meets the conditions, the basicity of sintering under each batching, the magnesium aluminum ratio, the element content, the sintering cost, and the comprehensive grade of charge after adding a certain lump ore. For example, taking the highest iron grade as the sintering batching optimization goal, the first five optimal sintering batching schemes are shown in Table 6. Taking the lowest sintering cost as the sintering batching optimization objective, the first five optimal sintering batching schemes are shown in Table 7.

**Table 6.** Optimal sintering batching scheme aiming at higher iron grade.

|  | Scheme 1 | Scheme 2 | Scheme 3 | Scheme 4 | Scheme 5 |
|---|---|---|---|---|---|
| Alkalinity | 1.247 | 1.247 | 1.243 | 1.241 | 1.238 |
| Mg/Al ratio | 0.617 | 0.602 | 0.611 | 0.601 | 0.605 |
| Fe (%) | 54.19 | 54.23 | 54.23 | 54.28 | 54.28 |
| Si (%) | 5.78 | 5.78 | 5.78 | 5.78 | 5.78 |
| Ca (%) | 11.49 | 11.49 | 11.44 | 11.42 | 11.4 |
| Mg (%) | 2.55 | 2.49 | 2.53 | 2.49 | 2.51 |
| Al (%) | 2.42 | 2.42 | 2.42 | 2.42 | 2.42 |
| Comprehensive grade | 56.245 | 56.279 | 56.28 | 56.315 | 56.316 |
| Sintering cost (¥) | 987.693 | 988.595 | 988.631 | 989.55 | 989.568 |
| Jinbuba (%) | 13.5 | 13.6 | 13.6 | 13.7 | 13.7 |
| 65 refined (%) | 20 | 20 | 20 | 20 | 20 |
| Super special (%) | 18 | 18 | 18 | 18 | 18 |
| FMG mixed (%) | 15 | 15 | 15 | 15 | 15 |
| Iron oxide scale (%) | 0.7 | 0.7 | 0.7 | 0.7 | 0.7 |
| Blast furnace dust (%) | 0.8 | 0.8 | 0.8 | 0.8 | 0.8 |
| High return (%) | 11.5 | 11.5 | 11.5 | 11.5 | 11.5 |
| Dust and mud (%) | 1.5 | 1.5 | 1.5 | 1.5 | 1.5 |
| Magnetic separation (%) | 1.2 | 1.2 | 1.2 | 1.2 | 1.2 |
| Rotary kiln lime (%) | 5.5 | 5.5 | 5.5 | 5.5 | 5.5 |
| Dolomite (%) | 6.3 | 6 | 6.2 | 6 | 6.1 |
| Limestone (%) | 2 | 2.2 | 2 | 2.1 | 2 |
| Self coking (%) | 4 | 4 | 4 | 4 | 4 |

**Table 7.** Optimal sintering batching scheme aimed at reducing sintering cost.

|  | Scheme 1 | Scheme 2 | Scheme 3 | Scheme 4 | Scheme 5 |
|---|---|---|---|---|---|
| Alkalinity | 1.329 | 1.332 | 1.334 | 1.334 | 1.339 |
| Mg/Al ratio | 0.663 | 0.659 | 0.655 | 0.669 | 0.661 |
| Fe (%) | 53.54 | 53.54 | 53.53 | 53.49 | 53.49 |
| Si (%) | 5.78 | 5.78 | 5.78 | 5.78 | 5.78 |
| Ca (%) | 12.26 | 12.29 | 12.31 | 12.31 | 12.35 |
| Mg (%) | 2.75 | 2.73 | 2.71 | 2.77 | 2.73 |
| Al (%) | 2.4 | 2.4 | 2.4 | 2.39 | 2.39 |
| Comprehensive grade | 55.743 | 55.742 | 55.742 | 55.707 | 55.706 |
| Sintering cost (¥) | 974.338 | 974.32 | 974.302 | 973.388 | 973.352 |
| Jinbuba (%) | 12.1 | 12.1 | 12.1 | 12 | 12 |
| 65 refined (%) | 20 | 20 | 20 | 20 | 20 |
| Super special (%) | 18 | 18 | 18 | 18 | 18 |
| FMG mixed (%) | 15 | 15 | 15 | 15 | 15 |
| Iron oxide scale (%) | 0.7 | 0.7 | 0.7 | 0.7 | 0.7 |
| Blast furnace dust (%) | 0.8 | 0.8 | 0.8 | 0.8 | 0.8 |
| High return (%) | 11.5 | 11.5 | 11.5 | 11.5 | 11.5 |
| Dust and mud (%) | 1.5 | 1.5 | 1.5 | 1.5 | 1.5 |
| Magnetic separation (%) | 1.2 | 1.2 | 1.2 | 1.2 | 1.2 |
| Rotary kiln lime (%) | 5.5 | 5.5 | 5.5 | 5.5 | 5.5 |
| Dolomite (%) | 6.9 | 6.8 | 6.7 | 7 | 6.8 |
| Limestone (%) | 2.8 | 2.9 | 3 | 2.8 | 3 |
| Self coking (%) | 4 | 4 | 4 | 4 | 4 |

Under the above batching restrictions, 2541 possible sintering batching schemes were generated by Python program and 87 sintering batching schemes meeting the requirements were screened, taking 6.05780 s. The program running environment is: central processing unit (Intel(R) Core(TM) i7-7700 CPU @ 3.60 GHz); memory (DDR4—4 GB).

It can be seen that the CP-IB automatic batching model can be competent for general sintering automatic batching operation and can calculate quickly and stably to produce

a sintering batching scheme and meet the needs of output and comparison of different sintering batching schemes.

## 4. Discussion

It is worth mentioning that the CP-IB algorithm originated from the research on intelligent sintering batching, which is applicable but not limited to the field of sintering batching. As the core algorithm of automatic batching, the CP-IB algorithm is applicable to all batching operations that need automatic batching and have a clear limit on the scope of raw material allocation.

In the field of industrial smelting, the CP-IB algorithm is applied to the sintering and iron ore powder batching process of pellets. Different batching schemes that meet the requirements can be obtained to meet the needs of comparing different batching schemes. At the same time, the CP-IB algorithm is applied to the preparation process of red mud, which helps to save on the cost of sintering [16]. In the field of food production [17,18], the CP-IB algorithm is used in the process of producing food ingredients, such as starch, yeast products, and seasonings, and a large number of food raw material ratio schemes can be obtained. Foods that taste better can be researched. The CP-IB algorithm is used in the manufacture of high-tech materials [19], such as the use of industrial silicon to smelt silica and the smelting of carbonaceous materials. The CP-IB algorithm can improve the accuracy of raw material allocation and produce materials of better quality. Applied to building materials [20,21], the CP-IB algorithm can be used in the batching process of silica, quartz sand, limestone, and other glass production and cement production, effectively improving the function of building materials. It is applied to the manufacture of daily necessities [22], such as the compounding of polyvinyl chloride resin particles in the plastisol industry, the compounding of rubber preparation of natural rubber, and the compounding of gel materials in the preparation of concrete, all of which can take advantage of the CP-IB algorithm. It can help in the manufacture of products that meet industry requirements and have better performance.

## 5. Conclusions

This paper proposes and implements a new automatic batching algorithm, the CP-IB algorithm. We have applied the CP-IB algorithm to our ongoing sintering batching research. We have solved the problem that the linear programming algorithm cannot be used in some production and the sintering batching relies too much on manual experience. It can help enterprises to complete the work of sintering ingredients and save on manpower and material costs. The algorithm has shown excellent results in the research.

**Author Contributions:** Conceptualization, methodology, and writing—original draft, X.R.; resources, visualization, and writing—review and editing, C.G. and H.W.; software and formal analysis, S.F.; data curation and validation, T.X.; investigation and project administration, A.Y. All authors have read and agreed to the published version of the manuscript.

**Funding:** This research was supported by the National Natural Science Foundation of China (NO. 52074126), Hebei Province Natural Science Fund for Distinguished Young Scholars (NO. E2020209082), Scientific Basic Research Projects (Natural Sciences) (NO. JQN2021027), and Hebei Natural Science Foundation Project (NO. E2021209024).

**Institutional Review Board Statement:** Not applicable.

**Informed Consent Statement:** Not applicable.

**Data Availability Statement:** The data used to support the findings of this study are available from the corresponding author upon request.

**Acknowledgments:** We thank the Hebei Intelligent Engineering Research Center of Iron Ore Optimization and Ironmaking Raw Materials Preparation Process and the Key Laboratory of Engineering Computing in Tangshan City of North China University of Science and Technology.

**Conflicts of Interest:** The authors declare no conflict of interest.

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
