# Peer review of "Automatic Ore Blending Optimization Algorithm for Sintering Based on the Cartesian Product"

_metals, doi:10.3390/met12081351_

Round 1

Reviewer 1 Report

Authors try to report 'Automatic Proportioning Model' for sintering of iron ore using the approach of cartesian technique which could reduce the amount of time spent in manual batching. Moreover, the reported work is alleged to be done first time. Before the work is publishable, I would strongly recommend following corrections:

1- Better, if the title of the manuscript is modified. 'Research' represents a broad term in the title and gives no clue on what actually the manuscript points to. Better to use terms like 'An interpolation algorithm' or 'A Systematic Investigation/model' or 'A Novel Optimization Method. etc. 

2-  Abstract is confusing. Rephrase. Authors describe first what they do and then later why it is needed, which is a wrong approach for audience. Better to convey first Where and Why it is needed and then How you plan to do. For eg. Line 24-28 should be shifted above and then follows Line 18-24.

3- Line no 21 (abstract). please rephrase 'Sintering Batching' 

4- At various places in the introduction the words are repeated more than twice in the same sentence. Eg. Line 32-34, authors use two times the word ' Ore Blending'. Please rephrase throughout such mistakes in Introduction part. 

5- What is the exact objective of the study? The title say authors try to do some research. The abstract say authors try to construct an automatic batching algorithm. In Introduction (Line 140-143) authors say they wish to develop an automatic proportioning model. Its better to use similar words in abstract, introduction and conclusion. For now, everything has some new method. Please unify and rephrase throughout the paper.

6- Please check the references. I cannot find any mentioned references if I make individual search on google. Do they have DOI? Are they published manuscript or book or magazines? 

7- I see that there are already few studies made on sintering optimization technique using cartesian approach? Why haven't authors cited them? 

8- Please ask a native speaker to check on english and grammer.

5- Line 142, authors report that the work is done first time. If so, it is better to write this in Abstract. This is the novelity in the work and should be said in the abstract and if possible in the title. 

6- Rephrase abstract and conclusion together. Currently, the conclusion is too too too confusing. Its very difficult to exactly get out the objective and results from the conclusion. 

7- Please remove any reference from the conclusion. and please rephrase it to be more concise, small and understandable. 

8- If you need to convey your idea. probably write another subsection as discussion or applicability area. In conclusion please try to state how your objectives are solved. 

Reviewer 2 Report

This research study is interesting for industrial applications. However, there are some suggestions before the publication of the article.

Manuscript ID: metals-1799090

Type: Article

Title: Research on automatic ore blending algorithm of sintering based on Cartesian product

1- There are so many subtitles under section 2.2 (2.2. CP-IB mathematical model), that it is suggested to collect them under the main title. Similar processes are there also in other sections.

2-In table3, the content of compositions is wt.% or not? Please indicate it. Additionally, it is unnecessary to write in each colon as “…. content”. It is clear that shows the content of the materials. Also, a similar correction is needed to be done in Table 4, and Table 5.

3- The unit of parameters in tables 6 and 7 are needed to be given.

Reviewer 3 Report

The contribution is well prepared by the authors and I am convinced that it will interest the readers.

I will ask the authors for the following corrections:

- image 1 is unreadable

- 2.3.2 algorithm is broken (line 219)

- Table 3, Table 4 - Scattered text

  After editing, I recommend for publication.

Round 2

Reviewer 1 Report

The authors have made impressive improvements. The manuscript is now very understandable and demonstrates the idea behind the work done. Hence, I recommend the publication of this manuscript. Thank you!